# Transformation of Agricultural Landscapes and Its Consequences for Natural Forests in Southern Myanmar within the Last 40 Years

Phyu Thaw Tun [ID], Thanh Thi Nguyen *[ID] and Andreas Buerkert [ID]

Organic Plant Production and Agroecosystems Research in the Tropics and Subtropics, Faculty of Organic Agricultural Sciences, University of Kassel, 37213 Witzenhausen, Germany; phyuthawtun@uni-kassel.de (P.T.T.); buerkert@uni-kassel.de (A.B.)
* Correspondence: tropcrops@uni-kassel.de; Tel.: +49-55-4298-1229

**Abstract:** Kyunsu township comprises coastal regions and a multitude of small islands covered by vast tropical evergreen and mangrove forests, and a large water body in the Adman Sea of Myanmar. Due to population growth, residents have increasingly expanded their agricultural land areas into natural tropical evergreen and mangrove forests, leading to deforestation. Understanding the processes and consequences of landscape transformation for surrounding ecosystems is crucial for local policy making and for fostering sustainable crop production in this area. Landsat datasets from 1978, 1989, 2000, 2011, and 2020 were used in a time-series post-classification approach to investigate land use land cover (LULC) changes in the Kyunsu township of Southern Myanmar across the last 40 years. Our study also attempted to assess the effects of the transformation of LULC on carbon stocks. Between 1978 and 2020, major LULC changes occurred with the expansion of Paddy Fields (+90%), Plantations (+11%), Open Forests (+81%), Settlement Areas (+115%), Aquaculture Areas (+1594%), and Others (+188%) while the area covered with Closed Forests shrunk by 44% and with Mangrove Forests by 9%. Water Bodies expanded by 0.13%. Our analyses show that between 1978 and 2020 2453 ha of Paddy Fields expanded into Plantations, 1857 ha to Open Forests, and 1146 ha to Mangrove Forests. Additionally, 12,135 ha of Open Forests, 8474 ha of Closed Forests, and 2317 ha of Mangrove Forests became Plantations. Across the 40 year study period, a total of 40,523 ha of Closed Forests were transformed to Open Forests. Our findings show that transformation of agricultural landscapes in the study area significantly affected deforestation and forest degradation of tropical evergreen rain forests and mangrove forests which are vital sources of ecosystem services. These transformations led to estimated losses of carbon stocks between 1978 and 2020 ranged from 89,260–5,106,820 Mg (average of 1,723,250 Mg) in our study area. Our findings call for sustainable resource intensification to increase production efficiency in existing cultivated areas rather than crop land expansion into natural forests. In addition, our data highlight the need for rigorous policies to conserve and protect tropical natural evergreen and mangrove forest, as key local resources providing multiple ecosystem services.

**Keywords:** crop expansion; deforestation; ecosystem services; forest degradation; Kyunsu township; tropical forests

## 1. Introduction

Myanmar's forests provide important resources for the livelihoods of its people and the economy of the nation. About 44% of the country's total land area is still occupied by natural forests [1] and most of them are under state ownership and managed as public land by the Forest Department [2]. Myanmar's forests are providers of timber, fuelwood, bamboo, honey, bee wax, and orchids [1]. Such forests are also known to be dynamic sources and sinks of greenhouse gases. However, securing environmental services such as

carbon storage, water regulation, and biodiversity conservation depends on sustainable forest management [3].

Myanmar's coastal regions are characterized by low-intensity rice (*Oryza sativa* L.) production, or remain covered by the natural vegetation of evergreen rainforests and mangrove forests [4]. It was recently estimated [5] that 81% of the Tanintharyi region was covered by intact forests, whereby 42% were upland evergreen forests, 22% lowland evergreen forests, 11% mixed deciduous forests, and 6% mangrove forests. Mangrove forests in coastal regions govern important food webs of aquatic organisms which support food and income for coastal communities, supply timber and fuel wood for the residents, and provide shoreline stabilization, reducing the impact of flooding during storms as their root mats protect the shoreline from erosion [6]. The coastline and archipelagos in the region are considerably vulnerable to the effects of storms and rises in sea level induced by global climate change [3]. Conservation and protection of mangroves as natural resources and buffer zones are crucial for the sustainable livelihoods of coastal communities.

Globally, deforestation is mainly driven by crop land expansion [7]. Approximately 60% of new agricultural land in Southeast Asia is derived from intact forests, and >30% from disturbed forests [8]. Due to rapid population growth and favorable climates for high value crops, forest areas in tropical countries are increasingly threatened by expansion of agricultural land. Additionally, Myanmar's population rose from 44.7 million in 1995–1996 [9] to 51.1 million in 2014 [3], leading to excessive use of ecosystem services and subsequent ecosystem degradation including reduction of carbon stocks for food demand [7]. Assessment of threats faced by different types of forests and planning strategies for forest conservation should be based on accurate information of forests and surrounding land cover distribution.

Time-series analysis of LULC changes using satellite datasets allows us to detect distribution, changes, and transformation of LULC within a given period [10]. It indicates threats to particularly valuable ecosystems and may thus be useful for land use planning [11]. Although several approaches to LULC classifications such as pixel-based, knowledge-based, contextual-based, object-based, and hybrid approaches [12] have been developed, the complexity and heterogeneity of landscape features, availability of remote sensing datasets, and limited effectivity of image processing and classification approaches may limit the success of accurate LULC classifications [13,14]. Many studies adopted hierarchical classifications, especially for time series in LULC analysis. Even if automated cartographic studies reduce human errors, speed up data processing, and provide accuracy of final outputs [15], it is the combination of automatized and manual approaches that may lead to improved classification accuracy in complex landscapes [12].

Kyunsu, a remote area in Tanintharyi region of Southern Myanmar, exhibits such a complex and heterogeneous landscape, as it comprises evergreen rain forests and mangrove forests. Rising demands for food and income from a growing population have driven the expansion of agricultural lands into natural forests, leading to losses of natural resources for ecosystem services. On the other hand, ecosystem protection programs in the area restrict agricultural land expansion. A solid understanding of the changes in and transformation of LULC and their potential consequences on ecosystems is crucial to support local policy-making that is geared towards fostering sustainable crop production in this area. While a previous study by Htwe et al. [16] has provided first insights into the land use patterns of the region, a systematic LULC analysis is lacking for this part of Southeast Asia.

Our study aimed at filling this knowledge gap, whereby its objectives were to (i) develop an approach to accurately classify LULC features of the Kyunsu township using remote sensing data of 1978, 1989, 2000, 2011, and 2020, followed by detecting the changes in and transformation of LULC, and (ii) assess the effects of the changes in and transformation of LULC on local ecosystem services within the last 40 years.

## 2. Materials and Methods

### 2.1. Description of the Study Area

Kyunsu township (Figure 1) is part of the Mergui district in the Tanintharyi region of Southern Myanmar. It is located between 11°25′03″N to 12°50′12″N and 97°13′48″E to 98°56′13″E at an elevation of 0–771 m above sea level (asl, [17]). It belongs to the Tanintharyi coastal region bordered by the Adman Sea (a part of the Bay of Bengal) and includes a large part of the Mergui Archipelago, which comprises more than 800 islands covering about 3,434,000 ha and up to 30 km offshore. Coral reefs surround the outer islands, and mangrove stands cover many of the inner islands. A total of 34,770 ha is protected as public evergreen rain forests and 25,900 ha as protected public mangrove forests [4]. In 2014, when Myanmar became the 11th "Mangrove for the Future" member country [18], Kyunsu township became a project site for mangrove forests' protection. The land area of Kyunsu township comprises 476,160 ha [19] with a total population of 171,750, of which 97% are rural residents [20]. Kyunsu town covers 2428 ha [19] with a population of 5548 [20]. A total of 15 out of 20 settlements are hard to reach given poor road infrastructure [21]. About 60% of the employed people in the township are farmers, forest users, and fishermen [20]. Local average annual rainfall is 2265 mm, with a mean temperature of 28 °C between 2012 and 2021 [22]. Based on geographical features and livelihood strategies, the area is divided into three agroecological zones: the sea zone, the lowland zone, and the plantation zone (Table 1, [19]).

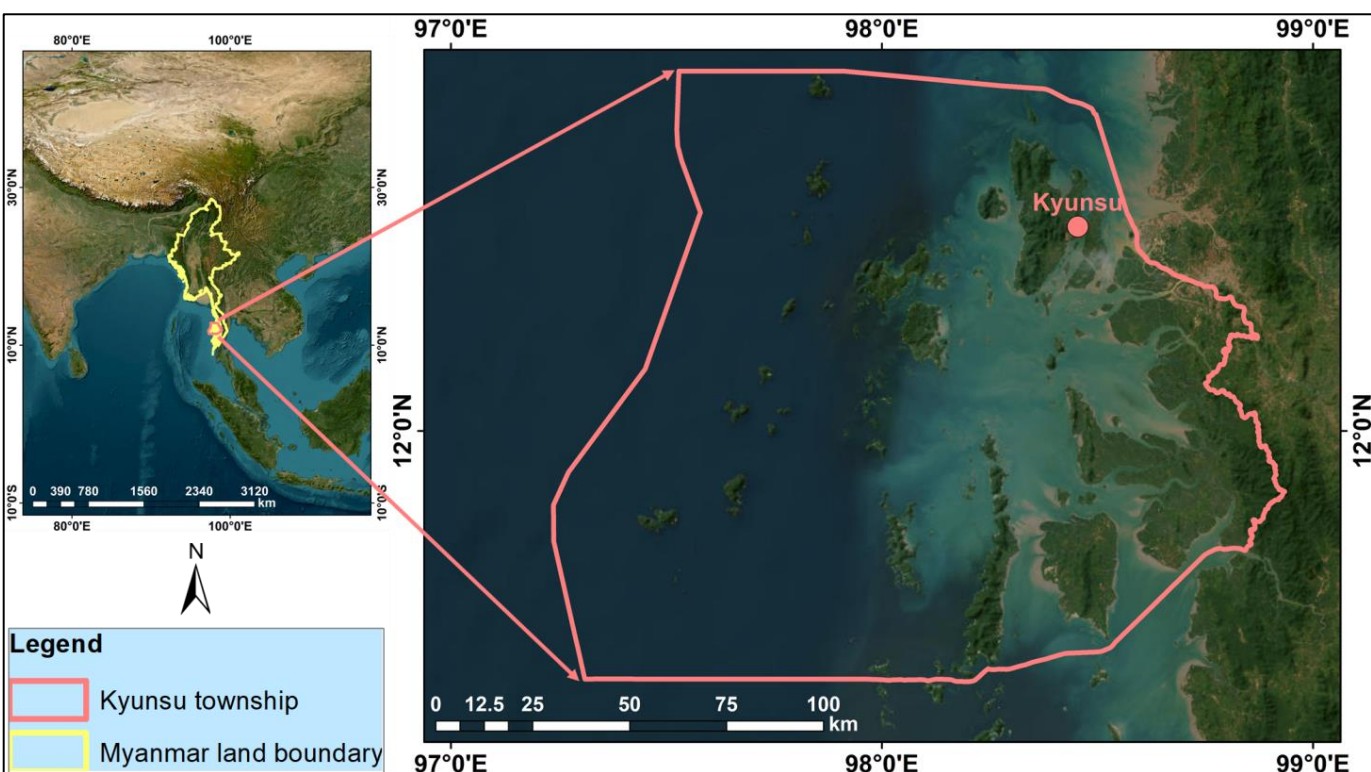

**Figure 1.** Map of Kyunsu township in Southern Myanmar ([23]; (ESRI, Maxar, GeoEye, Earthstar Geographic, CNES/Airbus DS, USDA, USGS, AeroGRID, IGN, and GIS User Community (2022). The boundary of the study area was digitized based on an Open Street Map using ArcGIS Desktop 10.4.1 developed by Environmental Systems Research Institute (ESRI) in 2016 [24].

### 2.2. Acquisition of Datasets for the Classification of LULC Features

Over the years, the United State Geological Survey (USGS) has released Landsat Collection 1 data which share common radiometric and geometric parameters. Collection 1 data have been designed as Tier 1 (T1) and Tier 2 (T2). Tier 1 data are recommended for

time-series analysis as the data have the highest radiometric and position quality and precision terrain processing, and have been inter-calibrated across the Landsat sensors. Tier 2 data also have good quality; however, radiometric calibration and collection of ground control points may be limited by cloud cover [25]. Use of data within the Landsat program allows the detection of long-term regional and global LULC changes because the data are sufficiently consistent with data from earlier remote sensing missions [26]. Our study used Landsat collection 1 Tier 1 and Tier 2 data for time series analysis of LULC. To cover our entire study area, four adjacent satellite datasets were required for each year. Four Landsat 3 Multispectral Scanner (MSS) Tier 2 datasets of 1978, three Landsat 5 Enhanced Thematic Mapper (ETM) surface reflectance Tier 1 datasets of 1989, 2000, and 2011 each, and three Landsat 8 Operational Land Imager (OLI)/Thermal Infrared Sensor (TIRS) surface reflectance Tier 1 datasets of 2020 (Table 2) were acquired from Google Earth Engine (GEE) using JavaScript [27]. For a portion of the study area, Tier 1 datasets were not provided by the GEE platform for 1989, 2000, 2011, and 2020. To fill this data gap, one Tier 2 dataset for each year 1989, 2000, 2011, and 2020 was downloaded from USGS Earth Explorer site (Table 2, [28]). Dates of data acquisition were filtered from January to December for each year, whereby we also selected the datasets for minimum cloud cover percentage of 0–7%.

**Table 1.** Geographical features and livelihood strategies of three agroecological zones of Kyunsu township in Southern Myanmar.

| Agroecological Zones | Geographical Features | Livelihood Strategies |
|---|---|---|
| Plantation zone | Rain forests, mangrove forests, and coastal areas | Rainfed plantations on hilly lands, non-farm activities (civil servants, casual workers, home business such as shops, production of local snacks, and fishery) |
| Lowland zone | Flat plains, rain forests, mangrove forests, and coastal areas | Rainfed lowland rice production on flat plains, rainfed plantations on hilly lands, non-farm activities (civil servants, casual workers, home business such as shops, production of local snacks, and fishery) |
| Sea zone | Flat plains, rain forests, mangrove forests, and coastal areas | Rainfed lowland rice production on flat plains, rainfed plantations on hilly lands, non-farm activities (civil servants, casual workers, home business such as shops, production of local snacks, and fishery) |

Source: [19].

**Table 2.** (**a**) Year of acquisition, satellite sensors, dataset providers, dates of acquisition, paths and rows (Landsat senses), and spatial (pixel size; m) and spectral resolutions (bands) of the satellite datasets used for land use and land cover (LULC) changes and transformation analysis in Kyunsu township of Southern Myanmar. (**b**) Description and wavelength (μm) of the bands of satellite sensors of Landsat 3, Landsat 5, and Landsat 8 datasets used for our study in Kyunsu township of Southern Myanmar [29].

| (a) | | | | | | |
|---|---|---|---|---|---|---|
| Year | Satellite Sensor | Dataset Provider | Date of Acquisition | Path/Row | Spatial Resolution (Pixel Size; m) | Spectral Resolution (Band) |
| 1978 | Landsat 3 MSS | USGS | 20 November 1978 ** | 140/51 | 60 | 4, 5, 6, 7 |
| | Landsat 3 MSS | USGS | 20 November 1978 ** | 140/52 | 60 | 4, 5, 6, 7 |
| | Landsat 3 MSS | USGS | 25 December 1978 ** | 139/51 | 60 | 4, 5, 6, 7 |
| | Landsat 3 MSS | USGS | 25 December 1978 ** | 139/52 | 60 | 4, 5, 6, 7 |

**Table 2.** *Cont.*

| (a) | | | | | | |
|---|---|---|---|---|---|---|
| Year | Satellite Sensor | Dataset Provider | Date of Acquisition | Path/Row | Spatial Resolution (Pixel Size; m) | Spectral Resolution (Band) |
| 1989 | Landsat 5 ETM | USGS | 12 December 1989 * | 131/51 | 30 | 1, 2, 3, 4, 5, 7 |
| | Landsat 5 ETM | USGS | 12 December 1989 ** | 131/52 | 30 | 1, 2, 3, 4, 5, 7 |
| | Landsat 5 ETM | USGS | 21 December 1989 * | 130/51 | 30 | 1, 2, 3, 4, 5, 7 |
| | Landsat 5 ETM | USGS | 21 December 1989 * | 130/52 | 30 | 1, 2, 3, 4, 5, 7 |
| 2000 | Landsat 5 ETM | USGS | 03 February 2000 * | 130/51 | 30 | 1, 2, 3, 4, 5, 7 |
| | Landsat 5 ETM | USGS | 03 February 2000 * | 130/52 | 30 | 1, 2, 3, 4, 5, 7 |
| | Landsat 5 ETM | USGS | 10 February 2000 * | 131/51 | 30 | 1, 2, 3, 4, 5, 7 |
| | Landsat 5 ETM | USGS | 10 February 2000 ** | 131/52 | 30 | 1, 2, 3, 4, 5, 7 |
| 2011 | Landsat 5 ETM | USGS | 01 February 2011 * | 130/51 | 30 | 1, 2, 3, 4, 5, 7 |
| | Landsat 5 ETM | USGS | 01 February 2011 * | 130/52 | 30 | 1, 2, 3, 4, 5, 7 |
| | Landsat 5 ETM | USGS | 08 February 2011 * | 131/51 | 30 | 1, 2, 3, 4, 5, 7 |
| | Landsat 5 ETM | USGS | 08 February 2011 ** | 131/52 | 30 | 1, 2, 3, 4, 5, 7 |
| 2020 | Landsat 8 OLI/TIRS | USGS | 15 November 2020 * | 131/51 | 30 | 2, 3, 4, 5, 6, 7 |
| | Landsat 8 OLI/TIRS | USGS | 15 November 2020 ** | 131/52 | 30 | 2, 3, 4, 5, 6, 7 |
| | Landsat 8 OLI/TIRS | USGS | 10 December 2020 * | 130/51 | 30 | 2, 3, 4, 5, 6, 7 |
| | Landsat 8 OLI/TIRS | USGS | 10 December 2020 * | 130/52 | 30 | 2, 3, 4, 5, 6, 7 |

| (b) | | | | |
|---|---|---|---|---|
| Satellite | Sensor | Band | Description of the Band | Wavelength (µm) |
| Landsat 3 | MSS | Band 4 | G (Green) | 0.5–0.6 |
| | | Band 5 | R (Red) | 0.6–0.7 |
| | | Band 6 | NIR (Near Infrared) 1 | 0.7–0.8 |
| | | Band 7 | NIR 2 | 0.8–1.1 |
| Landsat 5 | ETM | Band 1 | B (Blue) | 0.45–0.52 |
| | | Band 2 | G | 0.52–0.60 |
| | | Band 3 | R | 0.63–0.69 |
| | | Band 4 | NIR | 0.76–0.90 |
| | | Band 5 | SWIR (Shortwave Infrared) 1 | 1.55–1.75 |
| | | Band 7 | SWIR 2 | 2.08–2.35 |
| Landsat 8 | OLI/TIRS | Band 2 | B | 0.45–0.51 |
| | | Band 3 | G | 0.53–0.59 |
| | | Band 4 | R | 0.64–0.67 |
| | | Band 5 | NIR | 0.85–0.88 |
| | | Band 6 | SWIR 1 | 1.57–1.65 |
| | | Band 7 | SWIR 2 | 2.11–2.29 |

* Tier 1 (T1) dataset that met geometric and radiometric quality requirements, ** Tier 2 (T2) dataset that did not meet Tier 1 requirements but had good quality. Path = Descending orbit of the satellite, Row= Latitudinal center line of a frame of imagery. MSS = Multispectral Scanner, ETM = Enhanced Thematic Mapper, OLI = Operational Land Imager, TIRS = Thermal Infrared Sensor, USGS = United States Geological Survey.

Four datasets of 1978 were mosaicked to obtain a single dataset that covered the entire area. Similarly, we proceeded for three datasets of 1989, 2000, 2011, and 2020 each. The overlapping scenes of the adjacent datasets were assigned the median pixel values. From 1989 to 2020, the data types within each year were different given three Tier 1 datasets and one Tier 2 dataset. Therefore, each Tier 2 dataset of 1989, 2000, 2011, and 2020 was

classified separately to eliminate unforeseen errors and was later mosaicked with the classified datasets which covered the remaining three portions of the study area. Thus, a classified dataset for the entire study area was finally obtained for 1989, 2000, 2011, and 2020. Considering that Tier 2 datasets did not have any geometric distortion and almost the entire area contained a well-defined water body, no geometric and radiometric corrections were necessary in our case.

The spectral features of forests, plantations, and paddy fields were similar during the paddy growing season. Available Landsat datasets coincided with the post-harvesting time of paddy which allowed us to discriminate barren paddy fields against the evergreen vegetation areas of forests and plantations. Acquisition of satellite datasets for this time also offered the advantage of lower cloud cover of 0–7%. The dry season in Southeast Asia, which lasts from November to March, provides greater availability of cloud-free satellite imagery [5].

*2.3. Definition of LULC Classes*

A combination of the following three sources of data: (1) visual inspections of freely available Google Earth imagery procured from a variety of data providers with ≤0.65 m resolution and a range of image acquisition dates [5], (2) land cover datasets from the land cover portal developed by the Asian Disaster Preparedness Center (ADPC, [30]), and (3) a 30 m resolution Digital Elevation Model (DEM) provided by the National Aeronautics and Space Administration Shuttle Radar Topographic Mission (SRTM, [17]) allowed us to distinguish the following nine LULC classes in the study area: (1) Water Bodies, (2) Paddy Fields, (3) Open Forests, (4) Closed Forests, (5) Mangrove Forests, (6) Plantations, (7) Settlement Areas, (8) Aquaculture Areas, and (9) Others. Water Bodies comprised a sea water body and river networks. Cultivated barren lands for seasonal rice crops were defined as Paddy Fields. Forest areas with low density of forest trees spanning more than 0.5 ha with trees >5 m and ≥10–<40% canopy cover [1] were Open Forests, while forest areas which were covered by high density of forest trees and spanned more than 0.5 ha with trees >5 m and ≥40% canopy cover [1] were defined as Closed Forests. Mangrove Forests consist of naturally and artificially grown mangrove species. Naturally and commercially grown perennial crop lands were attributed to Plantations. All built-up areas were categorized as Settlement Areas, whereas ponds for raising aquatic animals were classified as aquaculture areas. The Others class included road networks, sandy beaches, and bare land areas (Figure 2).

*2.4. Classification of LULC from 1978 to 2020*

As shown previously [31–33], a combination of multisource data and two or more classifiers provides more accurately classified datasets than using a single one. Since the landscape features of our study area were complex and also heterogenous within each individual feature, we developed a hierarchical classification approach with multiple classification methods to overcome the limitations of the application of single classifier. Our approach consisted of four procedural steps which increase the accuracy of time-series analysis. In Step 1, an iso-cluster unsupervised classification tool was applied to classify Water Bodies, Paddy Fields, and Forests using Landsat datasets combined with a normalized difference vegetation index (NDVI), while in Step 2, a supervised random forest (RF) classifier allowed classification of Open Forests, Closed Forests, and Mangrove Forests using Landsat datasets integrating the NDVI, the normalized difference water index (NDWI), the normalized difference moisture index (NDMI), the modified normalized difference water index (MNDWI), and DEM data. Plantation datasets from the land cover portal of ADPC [30] supported the extraction of the Plantations class in Step 3, followed by reclassification based on inspection of Google Earth images after cross-checking with Landsat datasets. Settlement areas, Aquaculture Areas, and Others were digitized on Google Earth images in Step 4 (Figure 3).

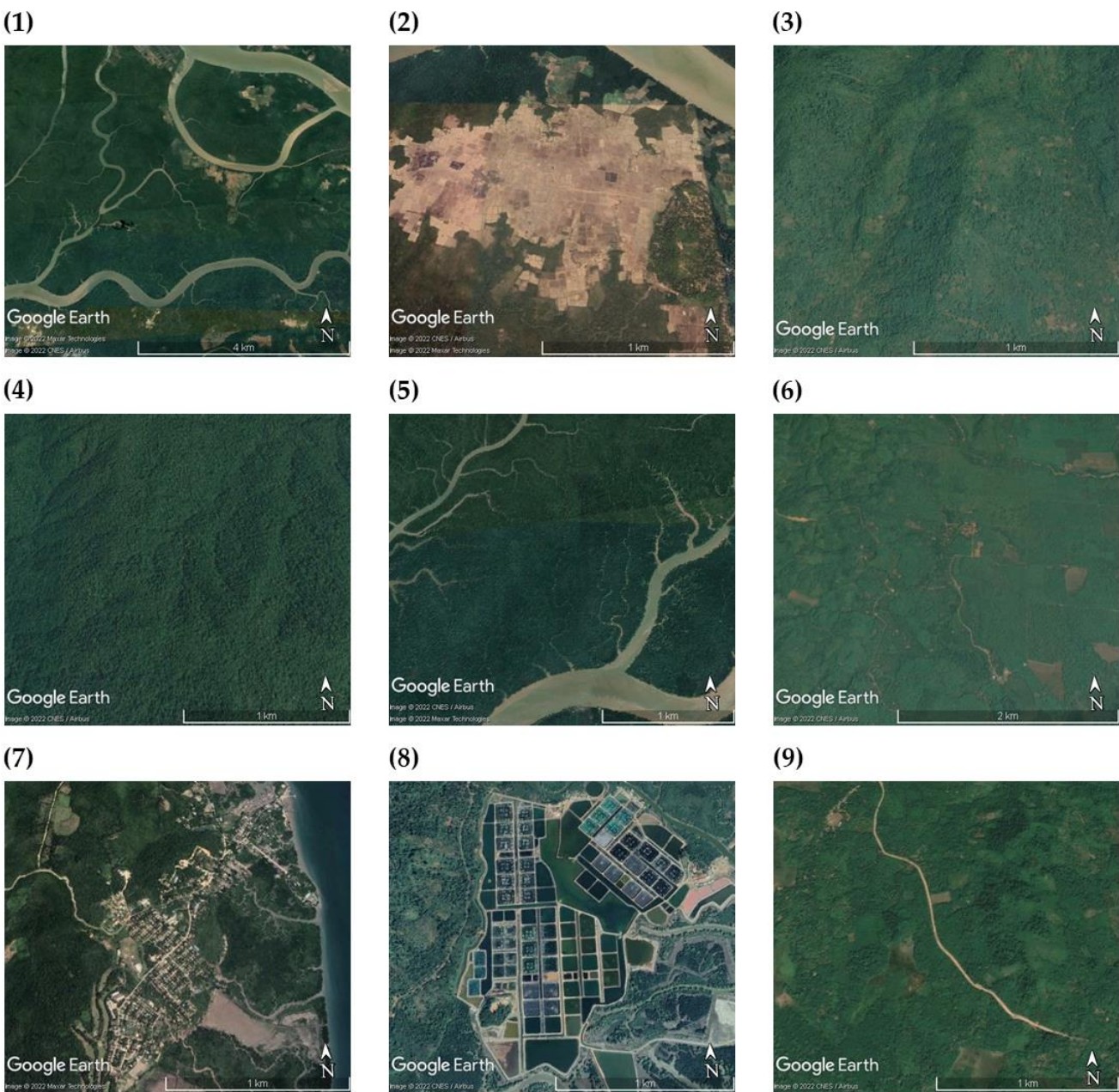

**Figure 2.** Land use and land cover (LULC) features/classes of Kyunsu township in Southern Myanmar representing (**1**) Water Bodies (river networks), (**2**) Paddy Fields (post-harvest condition without vegetation), (**3**) Open Forests, (**4**) Closed Forests, (**5**) Mangrove Forests, (**6**) Plantations, (**7**) Settlement Areas, (**8**) Aquaculture Areas, and (**9**) Others (road). Source: Google Earth Pro 7.1.

2.4.1. Iso-Cluster Unsupervised Classification (Step 1)

The study area is covered largely by Water Bodies and Forests. Paddy Fields were under non-vegetation post-harvest conditions during the data acquisition period. In this context, we applied the iso-cluster unsupervised classification tool in combination with NDVI data, which allowed us to initially classify Water Bodies, Paddy Fields, and Forests for the Landsat dataset of each year from 1978 to 2020 in ArcGIS. The method was also used for the classification of each T2 dataset of 1989, 2000, 2011, and 2020, as the dataset contained only Water Bodies and Forests. Unsupervised classification generates clusters based on similar spectral characteristics inherent in the satellite image/dataset [34]. Each cluster represents a specific LULC type [35]. As the spectral response of vegetation in the red band is highly related to chlorophyll concentration, which in the near infrared



band is dominated by leaf area index and green vegetation density [36], NDVI, defined as NIR-R/NIR+R [37], was used to maximize the contract between the greenness of forest vegetation, barren Paddy Fields, and Water Bodies. The NDVI was computed and combined with the selected bands (Table 2) of the Landsat datasets in Section 2.2 to obtain a dataset with NDVI data. This dataset was then used for the iso-cluster unsupervised classification to generate 50 initial classes. Upon inspection of the Landsat images by cross-checking with Google Earth images, these 50 classes were subsequently clustered by assigning the following three classes: Water Bodies, Paddy Fields, and Forests.

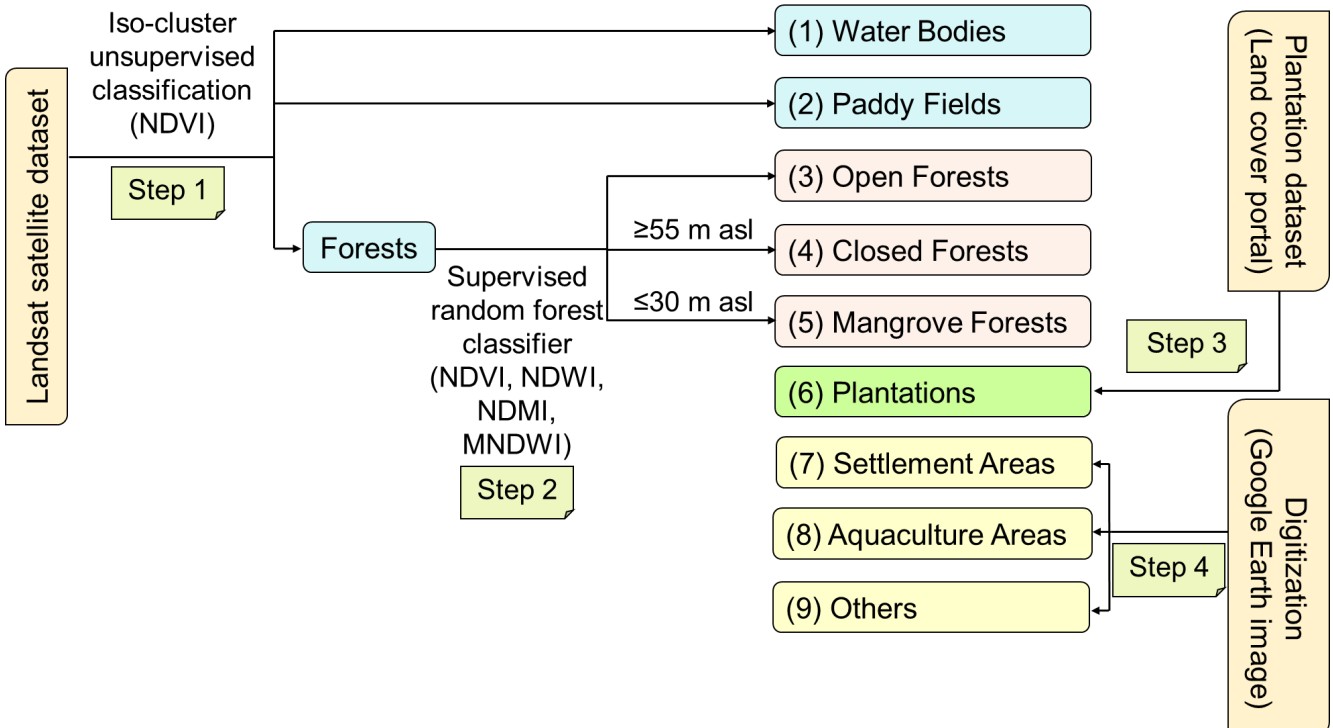

**Figure 3.** Flow diagram of procedural steps used in hierarchical classification of nine LULC features of Kyunsu township in Southern Myanmar.

### 2.4.2. Supervised Random Forest (RF) Classification (Step 2)

For forest classes, spectral confusion among Open Forests, Closed Forests, and Mangrove Forests posed a barrier to using an iso-cluster unsupervised classification for discrimination of those classes. This phenomenon was also reported by a study [5] which observed that the closed canopy forests, degraded forests, and young plantations of our study region were frequently misclassified based on Landsat imagery which resulted in a relatively low classification accuracy. To address this problem, Closed Forests and Mangrove Forests were classified separately using a supervised RF classifier [38] in GEE with the imported NDVI, NDWI, NDMI, MNDWI, and DEM layers as indicators. In a similar way, mangrove forests all over China were previously classified successfully [39].

To improve differentiation in Closed Forests' vegetation, the selected bands (Table 2) of the Landsat dataset in Section 2.2 were combined with NDVI data [40,41]. For calibration purposes, 115 training points and 298 training polygons for Closed Forests, and 380 training polygons for Non-closed forests (Water Bodies, Paddy Fields, Open Forests, Mangrove Forests, Plantations, Settlement Areas, Aquaculture Areas, and Others as listed in Table S1) were collected from Landsat datasets. Using these training data, a RF classifier was trained to classify Closed Forest and Non-closed Forest pixels. To reduce noise by removing single isolated misclassified pixels and further improve classification accuracy, the supervised classified Closed Forests dataset was post-processed using the Classified Sieve Tool of the Semi-Automatic Classification Plugin [42] in QGIS 3.20.2-Odense, developed by QGIS

Development Team in 2021 [43]. The size threshold 2 (single pixels were replaced by the value of the largest neighbor patch) and pixel connection 8 (diagonal pixels were also considered connected) were applied.

Topographic features such as elevation, slope, and aspect are known to influence distributions of land cover [41,44,45] and forests in hilly regions [13]. A recent study in Myanmar [5] has demonstrated that combination of medium-resolution multispectral Landsat datasets with topographic data can allow us to effectively differentiate between different types of tropical forests. Our study used the global DEM with 30 m resolution to evaluate the distribution of forest types in the study area and to extract forest classes, removing misclassified pixels. After identifying 50 inspection points of the DEM and Google Earth images, most Closed Forests were found at ≥55 m asl in this area. This was confirmed by another forest degradation analysis [46] in Southeast Asia, including Myanmar, which reported that remaining intact forests were concentrated in elevated inland areas. Therefore, areas ≥55 m asl of post-processed Closed Forests datasets were finally declared as such, whereas some forests at ≥55 m asl and forest areas <55 m asl became "Open Forests" and "Mangrove forests". The extracted Closed Forests dataset was then mosaicked with the unsupervised classified Forests class of Step 1 using ArcGIS.

Since Mangrove Forests are related to vegetation and water, the selected bands (Table 2) of the Landsat datasets in Section 2.2 were combined with the NDVI and NDWI, defined as G-NIR1/G+NIR1 [47] data for 1978, and NDVI, NDMI, defined as NIR-SWIR1/NIR+SWIR1 [48], and MNDWI, defined as G-SWIR1/G+SWIR1 [49] data for 1989, 2000, 2011, and 2020. Subsequently, the resulting datasets were used in the supervised RF algorithm for the classification of Mangrove Forests. For training purposes, 29 training points and 620 training polygons for Mangrove Forests, and 29 training points and 285 training polygons for Non-mangrove Forests (Water Bodies, Paddy Fields, Open Forests, Closed Forests, Plantations, Settlement Areas, Aquaculture Areas, and Others, Table S2) were selected from Landsat datasets. For the classification of Mangrove Forests, the same post-processing procedure as for Closed Forests was used.

After inspecting 50 points using the DEM and Google Earth images, most Mangrove Forests were identified at ≤30 m asl in our study area and classified as such. This was confirmed by a recent study [50] which also analyzed the distribution of Mangrove Forests in the inter-tidal zone of tropical and subtropical areas. The extracted Mangrove Forests dataset was mosaicked with the unsupervised classified Forests class of Step 1, which had been mosaicked with the extracted Closed Forests using ArcGIS. After extracting "Closed Forests" and "Mangrove Forests", the remaining forest areas assigned in Step 1 were classified as "Open Forests".

### 2.4.3. Referral and Reclassification of Classified Datasets (Step 3)

The pixels of Plantations were similar to those of forest classes, as most Plantations in the study area did not have a systematic planting pattern. A forest identification study in this region [5] has also observed that rapid expansion of plantation areas and large combined extents of bare land and degraded forest classes is a conservative indicator to estimate plantation areas. Clustering of Plantations with an iso-cluster unsupervised classification and collection of training samples to train a RF classifier were impossible in our study. Therefore, reference plantations data of 1987 (for 1978), 1989 (for 1989), 2000 (for 2000), 2011 (for 2011), and 2018 (for 2020) were obtained from the land cover portal of ADPC [30]. Upon inspection of Google Earth images, reference plantations data were reclassified in ArcGIS. Subsequently, the reclassified Plantations dataset was merged with the classified dataset of Step 2 using ArcGIS.

### 2.4.4. Digitization (Step 4)

The unclear spectral responses of Settlement Areas, Aquaculture Areas, and Others (road networks, sandy beaches, and bare lands) and heterogeneity within individual class limited the successful application of the iso-cluster unsupervised classification. The

noticeably small area contribution of these classes (0.03–0.06% of Settlement Areas, 0–0.06% of Aquaculture Areas, and 0.06–0.16% of Others), which were embedded in forest cover, hindered the collection of training samples to apply the supervised RF classifier. This agreed with the results of a recent review [51]. Therefore, Settlement Areas, Aquaculture Areas, and Others were digitized based on Google Earth images from the 1970s to 2020s by cross-checking with Landsat datasets of 1978, 1989, 2000, 2011, and 2020. A similar approach was reported previously [52]. The width of the road network was buffered at 10 m. The digitized vector datasets of these classes were transformed to raster datasets and consequently mosaicked with the classified dataset of Step 3 using ArcGIS.

*2.5. Post Classification Processing of Time Series Classified Data*

Independently classified datasets for 1989, 2000, 2011, and 2020 based on each Tier 2 dataset were mosaicked to obtain the classified dataset which covered the entire study area for each year. The classified dataset, which includes nine LULC classes, was clipped with the exact boundary (2,145,080 ha) of Kyunsu township using ArcGIS. To enhance classification accuracy, misclassified results of the clipped dataset of each year were reclassified using ArcGIS, based on the visual inspection of Google Earth images and Landsat datasets which were used in previous classification steps. The spatial resolution of the reclassified dataset was changed to 10 m to display the full road network in the Others class in the final classified dataset (Figure 4).

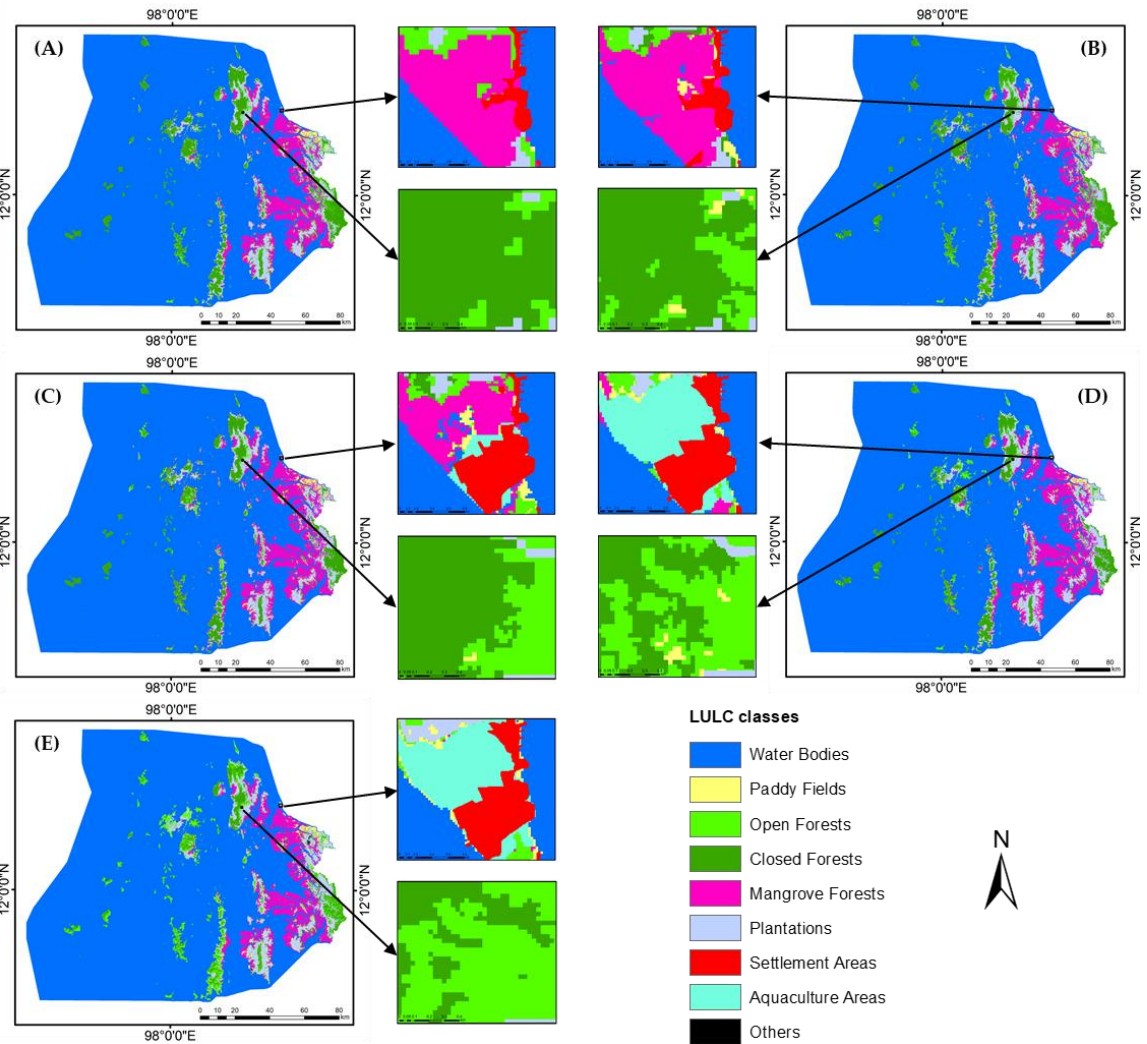

**Figure 4.** Final classified datasets of LULC classes in (**A**) 1978, (**B**) 1989, (**C**) 2000, (**D**) 2011, and (**E**) 2020 of Kyunsu township in Southern Myanmar.

### 2.6. Accuracy Assessment of the Final Classified LULC Datasets of 1978, 1989, 2000, 2011, and 2020

The accuracy of the final classified LULC dataset of each year was evaluated with a confusion matrix [53] of 900 random reference sample points generated by applying an equalized stratified random sampling strategy in ArcGIS. To achieve equal representation for the accuracy assessment, the same weight was assigned to each class (100 sample points per class). The actual LULC features of the reference sample points were visually identified on a Landsat image used for LULC classification by cross-checking with Google Earth images. Subsequently, the accuracy of the final classified dataset was determined by comparing the results of the final classified dataset of the reference sample points with the actual LULC features of the reference sample points using the confusion matrix. From the generated confusion matrix, the user's, producer's, and overall accuracies and the Kappa coefficient were determined.

### 2.7. Detection of Changes in and Transformation of LULC

To analyze the 40-year changes in and transformation of LULC in the area, the distribution (area in ha) of LULC classes was determined for each classified dataset. Changes between classified datasets were calculated [54] to understand their implications for ecosystem services and informed policy making.

### 2.8. Estimation of Carbon Stock Reduction Based on the Loss of Closed Forests and Mangrove Forests

We suspected a major reduction in carbon stocks in our study area as a consequence of the losses of Closed Forests and Mangrove Forests, because forests store a major proportion of carbon as above-ground biomass, thereby reducing the atmospheric $CO_2$ concentration [55]. Most forests in the study area are evergreen and mangrove forests [4]; therefore, the above-ground biomass data of evergreen and mangrove forests in the Southeast Asian region were reviewed from the literature, assuming similarities in forest structure and density across the region. The above-ground biomass of evergreen forests in Myanmar ranged from 1–152 Mg ha$^{-1}$, with an average of 49 Mg ha$^{-1}$. This range was computed from 88 sampling plots using national biomass expansion factor equations [56]. The estimated mangrove above-ground biomass in Vietnam ranged from 11–293 Mg ha$^{-1}$, with a mean value of 107 Mg ha$^{-1}$. This range was based on 121 sampling plots using the extreme gradient boosting decision tree algorithm [57]. The above-ground biomass of forests, including those of mangroves, depends largely on species composition and density [58]. Different topographies and climate conditions lead to major differences in the species composition and stand structure of Myanmar's forests [59]. Therefore, instead of an absolute value, a loss range of carbon stocks in our study area was computed for both Closed Forests and Mangrove Forests. Given the paucity of available data, estimations of the carbon stocks of other land use types were avoided. The proportion of carbon in the above-ground biomass of tropical forests is approximately 50% [60]. Hence, the total carbon stock loss was determined by multiplying the total biomass loss by 0.5.

The total carbon stock loss was calculated using an equation, modified after [61] as:

$$\text{TCSL} = [(\text{RCF} \times \text{BCF}) + (\text{RMF} \times \text{BMF}) - (\text{TCFMF} \times \text{BCF}) - (\text{TMFCF} \times \text{BMF})] \times 0.5$$

where TCSL refers to the estimated total carbon stock loss in megagram (Mg), RCF is the reduction in the area of Closed Forests in ha, BCF denotes the biomass of Closed Forests in Mg ha$^{-1}$, RMF is the area reduction of Mangrove Forests in ha, BMF is the biomass of Mangrove Forests in Mg ha$^{-1}$, TCFMF is the transformation area of Closed Forests to Mangrove Forests in ha, and TMFCF is the transformation area of Mangrove Forests to Closed Forests in ha. Forest changes between Closed Forests and Mangrove Forests were not considered to be forest losses. Therefore, area shifts between Closed Forests and Mangrove Forests were deduced in this calculation.

## 3. Results and Discussion

### 3.1. Accuracies of the Classified LULC Datasets

The hierarchical classification approach using multiple methods resulted in highly accurate classifications of LULC datasets. The overall accuracies of the final classified datasets varied between 96% and 97%, with Kappa coefficients of 0.96–0.97 (Table 3). The iso-cluster unsupervised classification tool with the integration of NDVI information allowed us to clearly differentiate Water Bodies, Paddy Fields, and Forests (Table 3). The RF classifier with the incorporation of NDVI, NDWI, NDMI, MNDWI, and DEM data allowed us to identify Closed Forests, Mangrove Forests, and Open Forests (Table 3). This confirms the potential of NDVI, NDMI, MNDWI, and DEM for the accurate identification of mangrove forests and other forest cover, as shown earlier [39,48]. The results also agree with another study [62] which reported that the combination of terrain data with remotely sensed data improved the accuracy in the classification of forest cover of Mount Hiei in Japan. The application of DEM data [63] led to an overall accuracy of 94% in mangrove classification. The reference datasets for Plantations from the land cover portal of ADPC [30] were successfully used to differentiate Plantations from Open Forests, Closed Forests, and Mangrove Forests. They allowed us to reclassify Plantations with high accuracies (Table 3). Digitalization of Settlement Areas, Aquaculture Areas, and Others on Google Earth images was carried out, and cross-checking with the Landsat datasets also confirmed the high classification accuracies (Table 3).

**Table 3.** User's accuracies (UA, %), producer's accuracies (PA, %), overall accuracies (OA, %), and Kappa coefficients (KC) of classified datasets for LULC classes: Water Bodies, Paddy Fields, Open Forests, Closed Forests, Mangrove Forests, Plantations, Settlement Areas, Aquaculture Areas, and Others of Kyunsu township, Southern Myanmar, in 1978, 1989, 2000, 2011, and 2020.

| LULC Classes | 1978 | | 1989 | | 2000 | | 2011 | | 2020 | |
|---|---|---|---|---|---|---|---|---|---|---|
| | OA (96.1) | KC (0.96) | OA (97.1) | KC (0.97) | OA (96.6) | KC (0.96) | OA (97.4) | KC (0.97) | OA (97.4) | KC (0.97) |
| | UA | PA | UA | PA | UA | PA | UA | PA | UA | PA |
| Water Bodies | 100 | 100 | 100 | 99 | 100 | 99 | 100 | 100 | 100 | 100 |
| Paddy Fields | 88 | 100 | 94 | 100 | 91 | 100 | 91 | 100 | 94 | 100 |
| Open Forests | 88 | 90 | 92 | 89 | 90 | 88 | 98 | 84 | 95 | 88 |
| Closed Forests | 93 | 87 | 93 | 95 | 94 | 91 | 92 | 97 | 91 | 98 |
| Mangrove Forests | 96 | 94 | 95 | 97 | 94 | 96 | 96 | 100 | 97 | 94 |
| Plantations | 100 | 99 | 100 | 100 | 100 | 100 | 100 | 100 | 100 | 100 |
| Settlement Areas | 100 | 100 | 100 | 100 | 100 | 100 | 100 | 99 | 100 | 99 |
| Aquaculture Areas | 100 | 100 | 100 | 100 | 100 | 100 | 100 | 100 | 100 | 100 |
| Others | 100 | 95 | 100 | 94 | 100 | 95 | 100 | 97 | 100 | 98 |

### 3.2. High-Performance Land Use Classification by Integrating Modern Techniques and Classical Approaches to Tackle Landscape Features

In our study, the combination of modern with classical approaches increased classification accuracy for the highly diverse landscapes. This was contributed to by the availability of datasets at the post-harvesting time of paddy to discriminate Paddy Fields from the evergreen vegetation areas of Forests and Plantations. The low cloud cover during the post-harvesting period facilitated visualization of local ground features during the dry season. In our study, the iso-cluster unsupervised classification failed to differentiate between Open Forests, Closed Forests, Mangrove Forests, and Plantations. The opportunity to select different types of training samples (points and polygons) on GEE facilitated collection of accurate ground truthing points to train RF as an effective classification tool for Closed Forest and Mangrove Forest pixels by integrating NDVI, NDWI, NDMI, MNDWI, and DEM. However, to enhance classification accuracy, misclassified pixels of Closed Forests and Mangrove Forests from RF classification were still to be reclassified by post-processing.

Additionally, it should be noted that in our study, Tier 2 datasets were utilized, whereby each sensor has been corrected with a specific algorithm [12].

Although we obtained reference datasets provided by ADPC [30] for Plantations, reclassification was necessary because we could not acquire exact datasets for the years of our time series. The small portions of Settlement Areas, Aquaculture Areas, and Others in our study area required us to digitize those classes based on Google Earth images between 1985 and 2020. Post-classification enhanced the accuracy of the final classified datasets to detect the changes in and transformation of LULC in the study area. Our work proved the benefits of multiple classification methods in LULC classification and the integration of multiple applications of GIS, and of remote sensing for the accurate LULC classification in regions with severe data limitations and heterogeneous landscape features.

### 3.3. Transformation of LULC within the Last 40 Years

Since most regions of the study area are close to the Adaman Sea, Water Bodies occupied the largest portion (82–83%) of the area between 1978 and 2020 (Figure 5). Water Bodies expanded constantly from 1978 until 2011, with a sharp increased between 1989 and 2000 (Figure 5). Comparison of the area of Water Bodies between 1978, 1989, and 2020, when the satellite datasets were acquired in the same months, showed an increasing trend. This was also observed between 1989 and 2020. This strengthens our findings on Water Bodies' expansion over 40 years, regardless of daily tidal variation. Between 2011 and 2020, however, the area of Water Bodies declined with the sharp regrowth of Mangrove Forests (Figures 6 and S2G, and Table S6). One cause for the initial expansion of Water Bodies could be the erosion of coastlines by strong waves from increasing coastal traffic of engine-powered marine transportation vehicles in combination with a rise in the sea level (Figure S1). Apparent regrowth of Mangrove Forests between 2011 and 2020 may also be the result of a major governmental project to protect mangroves in the study area.

As rice is the major staple food crop for Myanmar's population, it is not surprising that Paddy Fields in the study area expanded by 90%, as corroborated by a 115% increase in Settlement Areas between 1978 and 2020 (Figure 5). Paddy Fields mainly expanded into Plantations, Open Forests, and Mangrove Forests (Figures 6 and S2 and Table S7).

Across the study time period, Closed Forests continuously degraded to Open Forests (Figures 6 and S2 and Tables S3–S7) which increased by 81% (Figure 5). Myanmar's Forest Department [59] also reported a decrease in Closed Forests and an increase in Open Forests throughout the country from 2000 to 2015. This was also shown previously [64], whereby an expansion of Open Forests and other wooded land areas was noted, whereas Closed Forests significantly shrunk between 2005 and 2015. Nationwide, Open Forests increased up to 110% from 1989 to 2015 [1].

Across time, Closed Forests in the area shrunk by 44% (Figure 5). According to the global forest resource assessment report for Myanmar, from 1989 to 2015, nationwide Closed Forests declined by 61% [1]. Our study showed that the major factor contributing to the reduction of Closed Forests in the study area was the expansion of Open Forests and Plantations (Figures 6 and S2D,J and Tables S4 and S7). Our results agreed with a study [65] which showed that agricultural expansion including plantation development is a major cause of forest losses in Myanmar, and that closed canopy forests have experienced serious deforestation.

Excluding Water Bodies, Mangrove Forests were, with 34–40% of the total land area, the largest land cover class of Kyunsu township (Figure 5). Although the Tanintharyi region, especially the Mergui Archipelago, is considered the country's best conserved mangrove forest area, annual mangrove losses are estimated at 2.4%, whereas Myanmar's total mangrove forests declined by 13% from 2010 to 2015 [1]. Our study confirmed this phenomenon, showing a reduction in Mangrove Forests by 9% from 1978 to 2020 (Figure 5). Across the study period, large proportions of Mangrove Forests transformed into Water Bodies (Figures 6 and S2I, and Table S7) followed by Open Forests, whereas some Mangrove Forests transformed to Plantations, Paddy Fields, and Aquaculture Areas

(Figures 6 and S2J and Table S7). Although mangrove soils are not generally suitable for agriculture, agricultural expansion into mangrove forests to meet local food requirements is common in coastal regions of Myanmar [66]. Sizeable mangrove reforestation occurred in Plantations, especially between 1989 and 2011 (Figures 6 and S2D,F and Tables S4 and S5). Myanmar's Forest Department also launched a major reforestation and rehabilitation program in 2017 to restore degraded forests [59].

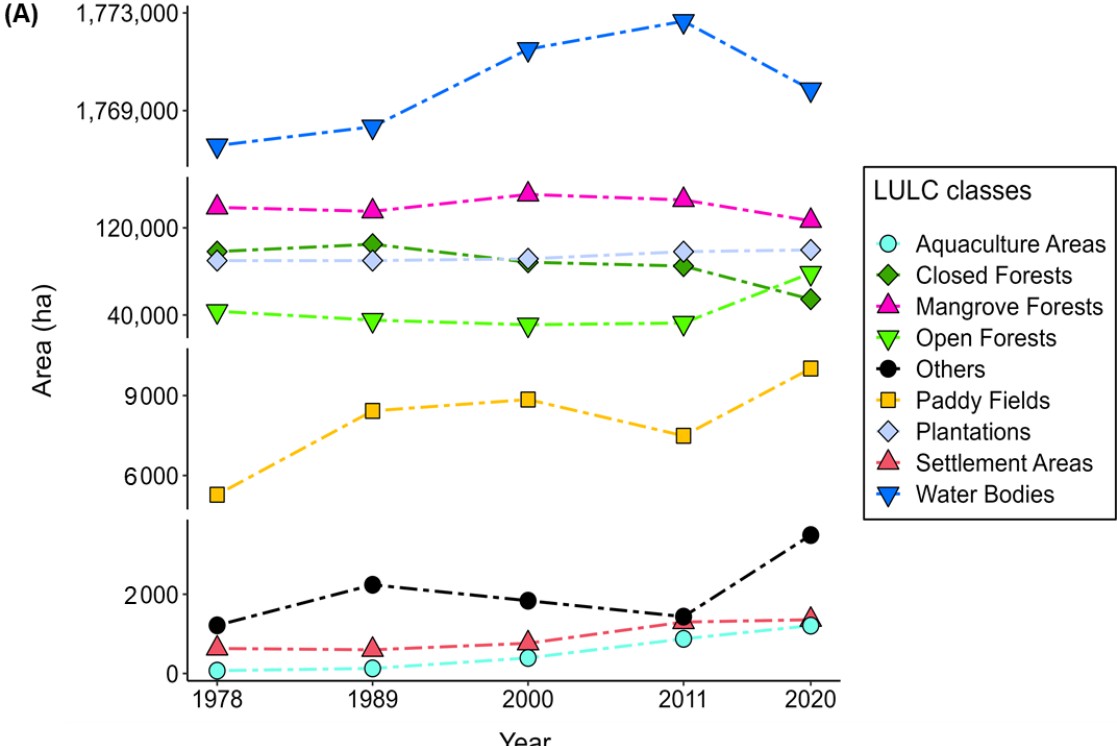

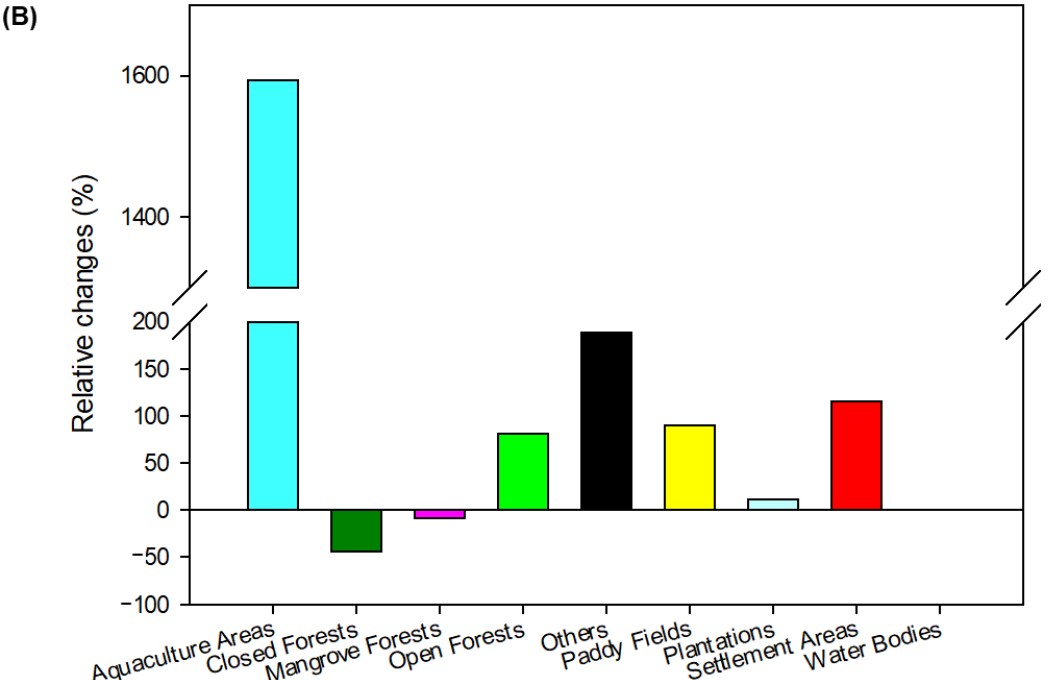

**Figure 5.** (**A**) Trends of the change in and (**B**) relative changes in LULC classes of Kyunsu township of Southern Myanmar across the last 40 years.

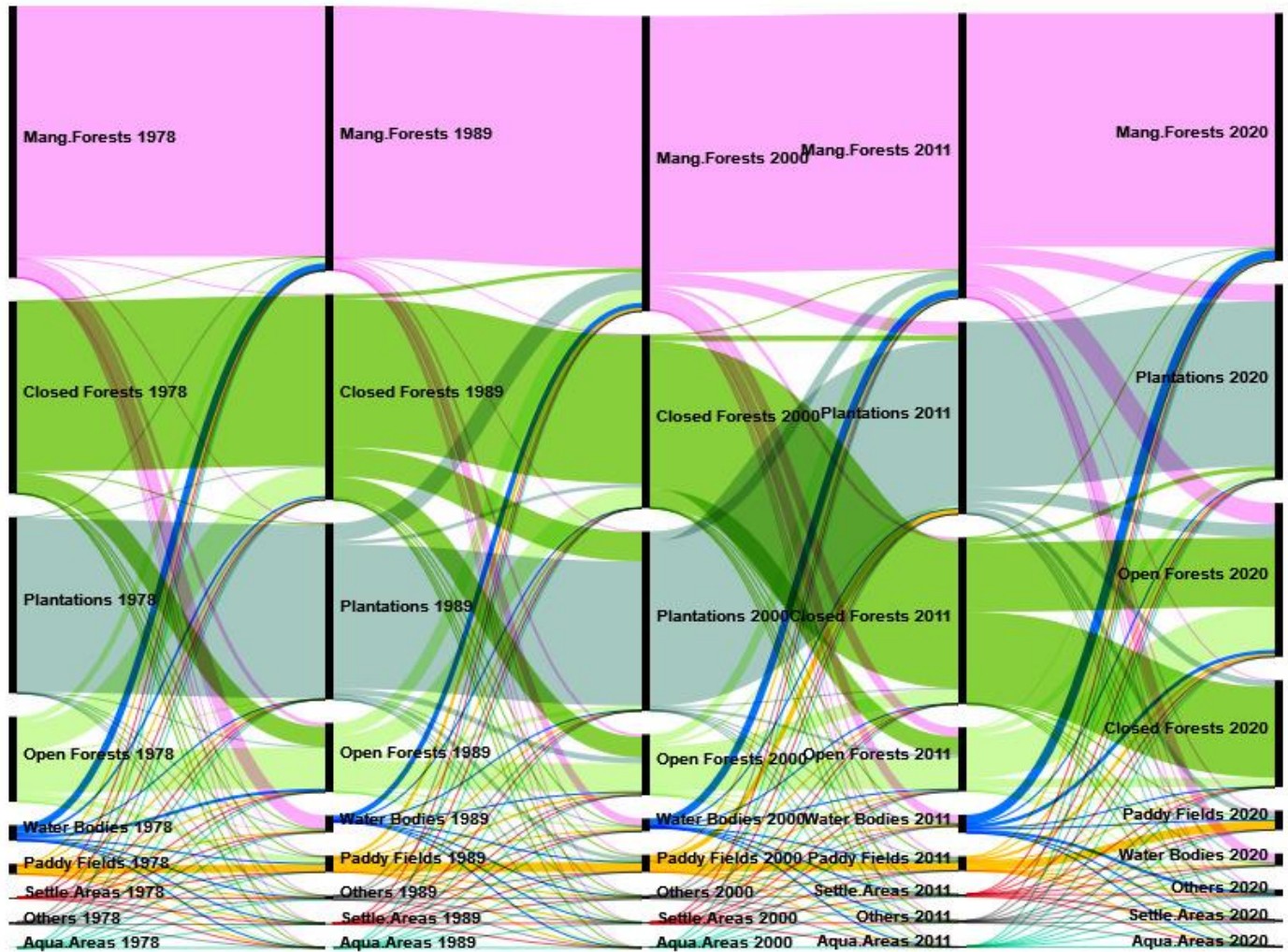

**Figure 6.** Transformation of LULC classes in Kyunsu township of Southern Myanmar across 40 years. The large area of stable Water Bodies in each year was excluded for the visualization of the transformation of the small LULC classes. Data of LULC transformation are provided in Tables S3–S7.

Our results showed the expansion of Plantations by 11% across 40 years (Figure 5), and the conversion of Open forests, Closed Forests, and Mangrove Forests to Plantations (Figures 6 and S2J and Table S7). Similar results were reported for the expansion of rubber (*Hevea brasiliensis* Müll.Arg.) plantations by 140% in the Mon and Tanintharyi regions from 1996 to 2007 [67].

Although our study area is only sparsely populated as compared to other regions of the country [6], Settlement Areas increased by 115% from 1978 to 2020 (Figure 5). It has been shown that settlement growth is an obvious indicator of population growth in many countries of the Global South [68], and this is also the case in our study area. This population increase mainly affected Plantations, Open Forests, and Mangrove Forests (Figures 6 and S2 and Table S7) as well as Aquaculture Areas (+1594%) and Others (+188%, Figure 5). The findings also reflect the rapid urbanization and industrialization of the country [3]. The land cover class most contributing to Aquaculture Areas were Mangrove Forests (Figures 6 and S2J and Table S7), reflecting the effects of the introduction of aquaculture to mangrove forests in 1980 [69]. A recent review [70] also showed that the development of aquaculture was one of many causes of losses in Myanmar's mangrove areas. The findings confirmed another report [71] that about 50% of global mangrove losses were associated with aquaculture. The environmental effects of future aquaculture projects

should thus be carefully assessed. Across 40 years, Water Bodies also contributed most to the transformation of Others (Figures 6 and S2 and Table S7).

### 3.4. Consequences of the Changes in and Transformation of LULC

Climate change-induced rises in global sea levels have been shown to cause erosion of coastlines, reduction of sandy beaches, and salt water intrusion in low lying crop fields in many parts of the world [72]. The expansion of sea water in our study area may have contributed to the abandonment of Paddy Fields near the coastlines [73]. Along with the growth of local settlements reflecting increasing population, rice producers extended their fields into Plantations, Open Forests, and Mangrove Forests (Figure 6) to meet local rice demand. Thus, the expansion of Paddy Fields was a considerable factor in the forest losses of Kyunsu township. Recent studies have shown that the expansion of Paddy Fields was the most important proximate driver of 47% (1996–2007, [67]), 68% (2007–2016, [67]), 88% (2000–2012, [74]), and 81% (1978–2011, [66]) of mangrove losses throughout Myanmar.

Forest degradation in Kyunsu township was likely aggravated by weak land use planning, extraction of timber, illegal logging, forest fires, expansion of subsistence agriculture and large-scale commercial plantations and agricultural lands, urbanization and infrastructure development, lack of alternative livelihoods, and widespread rural poverty [3]. Extraction of firewood and charcoal from the mangrove forests by coastal people was the main cause of the depletion of mangrove areas [69]. Plantations in the area transformed mainly into Open Forests, Closed Forests, and Mangrove Forests (Figure 6). The recent studies [75,76] have also pointed out that much of the forests' conversion to commercial agricultural land use occurred in the Tanintharyi region, whereby the rapid expansion of oil palm (*Elaeis guineensis* Jacq.) plantations and expansion of rubber plantations were the drivers of ongoing forest losses within the largest remaining areas of lowland evergreen forest in the Sundaic region of Southern Asia. Another cause of mangrove losses in this area is the rapid spread of aquaculture [74].

From 1978 to 2020, a large proportion of Closed Forests were transformed into Open Forests and other land use classes. This also happened to Mangrove Forests (Figure 6), reflecting the loss of large amounts of stored organic carbon and a subsequent decline in carbon sequestration by Plantations, Paddy Field and Water Bodies compared with forests. Terrestrial carbon stocks are strongly influenced by forest transformation to agricultural lands, the expansion of settlement areas and road networks, mining activities, and forest degradation [77]. Our study indicated that forest degradation and deforestation contributes to sizeable carbon emissions also in Myanmar [3,7]. We estimated losses in carbon stocks in our study area to total 89,260–5,106,820 Mg over the last 40 years, with an average of 1,723,250 Mg. This is the result of a reduction of 43,543 ha in Closed Forests and 12,270 ha in Mangrove Forests (Figure 5). In line with our findings, the NASA Jet Propulsion Laboratory estimated the above-ground biomass of closed forests and mangrove forests to range from 0 to 400 Mg ha$^{-1}$ by using boosting tree machine learning model [78]. The wide range of biomass estimations reflects the diversity of species in the tropical forests, but also different tree densities in closed forests and mangrove forests [56–58].

## 4. Conclusions

Our study demonstrates that a hierarchical classification approach using multiple methods allows to accurately determine different LULC classes in areas with severe data limitations and heterogeneous landscape features. In addition to assessment of the transformation of the agricultural landscapes, the results also highlight the effects of deforestation and forest degradation on ecosystem services. Over the last 40 years, many Open Forests, Closed Forests, and Mangrove Forests were transformed into Paddy Fields and Plantations, leading to losses of vital ecosystem services and of carbon stocks. The expansion of the sea water body and decreasing density of mangrove forests, along with erosion of coastlines, may cause the intrusion of seawater into low-lying paddy fields. Salt damage by coastal flooding as a consequence of the expansion of the sea water body could contribute to the

abandonment of paddy fields behind the coastlines. As rice is Myanmar's major staple food crop, losses of paddy fields along with limitations on the expansion of rice-cultivated areas into forest cover may reduce the security of staple food provision for local people. Although plantation agriculture is dominant in the area, expansion of plantations seems to be limited by forest protection programs which affect local livelihoods that rely on plantation agriculture. Our results call for the sustainable intensification of production in existing agricultural areas, rather than further crop expansion into natural forests which are vital for the provision of multiple ecosystem services to residents. Finally, our findings of deforestation and forest degradation in the area highlight the need for rigorous spatial planning to support natural forest conservation efforts.

**Supplementary Materials:** The following supporting information can be downloaded at: https://www.mdpi.com/article/10.3390/rs15061537/s1, Figure S1: Erosion of coastlines (B), (C), and (D) by the strong waves (A) and (C) from high engine-power marine transportation vehicles in Kyunsu township of Southern Myanmar in 2021; Figure S2: Transformation of LULC classes: Water Bodies (C1), Paddy Fields (C2), Open Forests (C3), Closed Forests (C4), Mangrove Forests (C5), Plantations (C6), Settlement Areas (C7), Aquaculture Areas (C8), and Others (C9) of Kyunsu township in Southern Myanmar across 40 years: (A) between 1978–1989 including Water Bodies, (B) between 1978-1989 excluding Water Bodies, (C) between 1989–2000 including Water Bodies, (D) between 1989–2000 excluding Water Bodies, (E) between 2000–2011 including Water Bodies, (F) between 2000–2011 excluding Water Bodies, (G) between 2011–2020 including Water Bodies, (H) between 2011–2020 excluding Water Bodies, (I) between 1978–2020 including Water Bodies, (J) between 1978–2020 excluding Water Bodies. Ticks indicate the area of LULC in 100,000 ha; Table S1: Reference training samples used for supervised random forest (RF) classifier to classify Closed Forests and Non-Closed Forests of Kyunsu township in Southern Myanmar across 40 years; Table S2: Reference training samples used for supervised RF classifier to classify Mangrove Forests and Non-Mangrove Forests of Kyunsu township in Southern Myanmar across 40 years; Table S3: Transformation matrix of LULC classes: Water Bodies (C1), Paddy Fields (C2), Open Forests (C3), Closed Forests (C4), Mangrove Forests (C5), Plantations (C6), Settlement Areas (C7), Aquaculture Areas (C8), and Others (C9) (in ha) of Kyunsu township in Southern Myanmar between 1978 and 1989; Table S4: Transformation matrix of LULC classes: Water Bodies (C1), Paddy Fields (C2), Open Forests (C3), Closed Forests (C4), Mangrove Forests (C5), Plantations (C6), Settlement Areas (C7), Aquaculture Areas (C8), and Others (C9) (in ha) of Kyunsu township in Southern Myanmar between 1989 and 2000; Table S5: Transformation matrix of LULC classes: Water Bodies (C1), Paddy Fields (C2), Open Forests (C3), Closed Forests (C4), Mangrove Forests (C5), Plantations (C6), Settlement Areas (C7), Aquaculture Areas (C8), and Others (C9) (in ha) of Kyunsu township in Southern Myanmar between 2000 and 2011; Table S6: Transformation matrix of LULC classes: Water Bodies (C1), Paddy Fields (C2), Open Forests (C3), Closed Forests (C4), Mangrove Forests (C5), Plantations (C6), Settlement Areas (C7), Aquaculture Areas (C8), and Others (C9) (in ha) of Kyunsu township in Southern Myanmar between 2011 and 2020; Table S7: Transformation matrix of LULC classes: Water Bodies (C1), Paddy Fields (C2), Open Forests (C3), Closed Forests (C4), Mangrove Forests (C5), Plantations (C6), Settlement Areas (C7), Aquaculture Areas (C8), and Others (C9) (in ha) of Kyunsu township in Southern Myanmar between 1978 and 2020.

**Author Contributions:** Conceptualization, P.T.T. and A.B.; methodology, P.T.T. and T.T.N.; software, P.T.T. and T.T.N.; validation, P.T.T., T.T.N. and A.B.; formal analysis, P.T.T. and T.T.N.; investigation, P.T.T. and T.T.N.; resources, P.T.T.; data curation, P.T.T.; writing—original draft preparation, P.T.T.; writing—review and editing, P.T.T., T.T.N. and A.B.; visualization, P.T.T. and T.T.N.; supervision, A.B.; project administration, A.B.; funding acquisition, P.T.T. All authors have read and agreed to the published version of the manuscript.

**Funding:** This study was funded by the German Academic Exchange Service (DAAD) through a scholarship to the first author with personal reference number: 91729808 under the funding programme/-ID: Research Grants—Doctoral Programmes in Germany, 2019/20 (57440921). We acknowledge funding from the University of Kassel (Germany) for the second author.

**Data Availability Statement:** Not applicable.

**Conflicts of Interest:** The authors declare no conflict of interest.

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
