# Peer review of "Transformation of Agricultural Landscapes and Its Consequences for Natural Forests in Southern Myanmar within the Last 40 Years"

_remotesensing, doi:10.3390/rs15061537_

Round 1

Reviewer 1 Report

This work is excellently written in a scientific sense, where the authors, in addition to the transformation of land cover and land use, connect with other different scientific areas, such as erosion caused by the action of sea waves, ecosystem services for the needs of the population, the application of multifunctional classification and application remote sensing.

I suggest that this paper be accepted as it is, because the authors have submitted a large amount of scientific papers dealing with this topic and issue. My suggestion is very short and consists of a few comments.

Figure 1 should be corrected a little, if you can see and expand the view in Myanmar boundary located, to see the wider location with other countries in Asia. You don't need to see all of Asia, but at least a part of the continent.

I suggest replacing the term "lies" with some other term such as “spreads”, “is located” or some other (line 85)

Title 2.2. Acquisition of Landsat datasets for the classification of LULC features to be on the same page as the text of this title

The text on line 134 and 135 if it can be on the same page as table 2 itself

Table of Contents line 638-661 should be deleted, it should not be in the paper.

Reviewer 2 Report

My main concern is that the majority of the manuscript is allocated to quantifying the LULC changes, and only in the last part of the discussion, the consequences of these changes were slightly discussed. I suggest that the authors improve their discussion in this part.

other comments:

The title is very long.  I suggest making it shorter.

Line 21-23 what is the base year of these relative changes? In other words, what is the timeline when for example the authors stated 90% expansion of paddy rice fields?

The abstract does not provide any information about the impacts of LULC changes while it is one of the main focuses of the study.

Which criteria were used to distinguish between the different types of forests? and what was the rationale for selecting such criteria?

The 2.8. section (Estimation of carbon stock reduction), is not clear. I suggest re-writing this section with more accurate language.

I suggest adding a subplot to Figure 5, that shows the relative changes as well.

Reviewer 3 Report

The manuscript aims a hierarchical classification of satellite images, also performing a temporal analysis of Kyunsu township, part of the Mergui district in the Tanintharyi region of Southern Myanmar.

Below the opinion divided into major and minor reviews

Major reviews

Abstract

Rewrite, contextualize first, then discuss the methodology used and the results (do not mention all the changes between classes)

Methodology

In studies involving remote sensing, the iso-cluster unsupervised classification technique is the largest use. What new scientific advancement does the manuscript make in the field of image classification? The recent developments in this field show a greater availability of more modern techniques, such as the use of machine learning, decision trees, or artificial neural networks that are widely used. 

I suggest that the authors reformulate their image classification methodology or provide well-founded justifications that demonstrate the advances achieved with the use of hierarchical classification, within a context where this technique has already been overcome by significant advances in the area of remote sensing. Below are some suggestions for articles dealing with image classification:

Asokan, A.; Anitha, J.; Ciobanu, M.; Gabor, A.; Naaji, A.; Hemanth, D.J. Image Processing Techniques for Analysis of Satellite Images for Historical Maps Classification—An Overview. Appl. Sci. 2020, 10, 4207. https://doi.org/10.3390/app10124207

Miao Li, Shuying Zang, Bing Zhang, Shanshan Li & Changshan Wu (2014) A Review of Remote Sensing Image Classification Techniques: the Role of Spatio-contextual Information, European Journal of Remote Sensing, 47:1, 389-411, DOI: 10.5721/EuJRS20144723

D. Lu & Q. Weng (2007) A survey of image classification methods and techniques for improving classification performance, International Journal of Remote Sensing, 28:5, 823-870, DOI: 10.1080/01431160600746456

Mahmon, N. A. and Ya'acob, N. A review on classification of satellite image using Artificial Neural Network (ANN). 2014 IEEE 5th Control and System Graduate Research Colloquium, 2014, pp. 153-157, doi: 10.1109/ICSGRC.2014.6908713

Anil B. Gavade & Vijay S. Rajpurohit (2021) Systematic analysis of satellite image-based land cover classification techniques: literature review and challenges, International Journal of Computers and Applications, 43:6, 514-523, DOI: 10.1080/1206212X.2019.1573946

M. Sheykhmousa, M. Mahdianpari, H. Ghanbari, F. Mohammadimanesh, P. Ghamisi and S. Homayouni, "Support Vector Machine Versus Random Forest for Remote Sensing Image Classification: A Meta-Analysis and Systematic Review," in IEEE Journal of Selected Topics in Applied Earth Observations and Remote Sensing, vol. 13, pp. 6308-6325, 2020, doi: 10.1109/JSTARS.2020.3026724.

The multiple classification methods need a solid foundation. Different procedures could produce classification results that aren't what was expected.

The authors contend:

lines 112-115: For the classification of LULC in the study area, satellite datasets of 1978, 1989, 2000, 2011, and 2020 (Table 2) were acquired from the U.S. Geological Survey’s Earth Resources Observation and Science (USGS/EROS) Landsat sensors directly and via Google Earth Engine (GEE) using JavaScript [22]. 

The entire classification process could be completed using the GEE platform, which offers effective RF-based classifiers and other machine learning-based approaches, requiring programming skills.

It is not a good strategy to use GIS applications like ArcGIS and QGIS, which are not the best software for image classification processes.

The classification should be followed by a clearer explanation of how the NDVI, NDWI, NDMI, and MNDWI spectral indices were used, along with a definition of each index's purpose and how it was used in the approach. Here are some ideas for spectral index studies:

 A. Bannari, D. Morin, F. Bonn & A. R. Huete (1995) A review of vegetation indices, Remote Sensing Reviews, 13:1-2, 95-120, DOI: 10.1080/02757259509532298

Jinru Xue and Baofeng Su (2017). Significant Remote Sensing Vegetation Indices:

A Review of Developments and Applications. Journal of Sensors. Volume 2017, Article ID 1353691, 17 pages. https://doi.org/10.1155/2017/1353691

Minor review

lines 141-144: …allowed to distinguish the following nine land cover classes in the study area: (1) Water Bodies, (2) Paddy Fields, (3) Open Forests, (4) Closed Forests, (5) Mangrove Forests, (6) Plantations, (7) Settlement Areas, (8) Aqua culture Areas, and (9) Others.

  • Classes were delineated in figure 2 with letters and in the text with numbers. Should uniforms

table 3: replace class codes (C1, C2, C3…) with class names

lines 305-308: The total carbon stock loss was calculated using the equation modified from the equation of previous study [48] as: [(Reduction areaClosed Forests×BiomassClosed Forests)+(Reduction areaMangrove Forests×BiomassMangrove Forests)-(Transformation areaClosed Forests to Mangrove Forests×BiomassClosed Forests)-(Transformation areaMangrove Forests to Closed Forests×BiomassMangrove Forests)]×0.5.

  • insert into the formula pattern

I recommend acceptance after major revision of the manuscript.

Reviewer 4 Report

The Reviewer's comments are in the attached PDF file.

Round 2

Reviewer 2 Report

The authors made substantial improvements in the manuscript. 

I suggest that the authors add the limitations and possible biases of their approach in the abstract and/or conclusion. Example includes the differences in the spectral ranges of the different Landsat satellites used in their analysis, and how these may affect the outcome.

Additionally, I made a couple of minor comments in the abstract and introduction which can be find in the annotated PDF. 

Reviewer 3 Report

The authors made the requested changes. I support the publication of this version of the paper.

Reviewer 4 Report

The Reviewer's comment (after R2) is in the attached PDF file.
